# Combining Mass Spectrometry-Based Phosphoproteomics with a Network-Based Approach to Reveal FLT3-Dependent Mechanisms of Chemoresistance

**DOI:** 10.3390/proteomes9020019

**Published:** 2021-04-27

**Authors:** Giusj Monia Pugliese, Sara Latini, Giorgia Massacci, Livia Perfetto, Francesca Sacco

**Affiliations:** 1Department of Biology, University of Rome Tor Vergata, Via delle Ricerca Scientifica 1, 00133 Rome, Italy; moniapugliese@gmail.com (G.M.P.); saralatini207@gmail.com (S.L.); giorgiamassacci@hotmail.it (G.M.); 2Fondazione Human Technopole, Department of Biology, Via Cristina Belgioioso 171, 20157 Milan, Italy; livia.perfetto@fht.org

**Keywords:** AML, FLT3, drug-resistance, (phospho)proteomic, signaling-network, logic-model

## Abstract

FLT3 mutations are the most frequently identified genetic alterations in acute myeloid leukemia (AML) and are associated with poor clinical outcome, relapse and chemotherapeutic resistance. Elucidating the molecular mechanisms underlying FLT3-dependent pathogenesis and drug resistance is a crucial goal of biomedical research. Given the complexity and intricacy of protein signaling networks, deciphering the molecular basis of FLT3-driven drug resistance requires a systems approach. Here we discuss how the recent advances in mass spectrometry (MS)-based (phospho) proteomics and multiparametric analysis accompanied by emerging computational approaches offer a platform to obtain and systematically analyze cell-specific signaling networks and to identify new potential therapeutic targets.

## 1. Introduction

Over the last decades, molecular biology has witnessed tremendous technological and scientific breakthroughs. This progress has advanced our understanding of cell molecular functioning. We are now aware that signals sensed by a cell do not necessarily propagate in linearly. Signaling networks are rather assemblies of intricate highly connected modules controlling key biological processes in a context-dependent manner. It has also become clear that diseases are often due to perturbation of molecular networks, rather than consequences of single or few gene dysfunctions. In this context, the analysis of networks rewiring in healthy and diseased conditions is critical to understand the properties of the system and to identify new therapeutic strategies. However, because of their complexity and size, the dynamic characterization of signaling networks in health and during disease onset and progression often turns out to be a daunting task [1,2,3].

Different computational strategies have been developed to model signaling networks and predict their behavior upon environmental or genetic external or internal (that is, genetic) perturbations. Most of these approaches take advantage of high throughput proteomic or transcriptomic data. Although both data-types are necessary to build models faithfully mimicking a system behavior, mass spectrometry (MS)-based proteomics is unique for this scope as it enables to measure both the concentrations and the activation levels of thousands of proteins in different conditions [4]. Over recent years, MS-based proteomics has undergone dramatic advances in sample preparation, instrumentation and computational methods. Thanks to these developments, it is now possible to quantify global changes in proteins and in post-translational modifications between different cellular states at great depth, enabling to address a biological complex [5,6]. Very recently, a robust workflow combining advances at multiple levels, including sample preparation, liquid chromatography and mass spectrometer, enabled to accurately and robustly quantify proteomes and their changes in single, FACS-isolated cells [7]. Altogether these observations show that mass spectrometry-based proteomics is ready for new challenges, addressing health-relevant questions. Here we describe how the combination of MS-based (phospho)proteomics with literature-derived signaling networks may help to identify new therapeutic targets to revert the chemoresistance of FLT3-dependent acute myeloid leukemia.

Acute myeloid leukemia (AML) is characterized by a dysfunction of the hematopoietic process resulting in the perturbation of the balance between stem cell proliferation and differentiation. In the last decade, genomic studies identified many different genetic alterations in AML patients, revealing the complexity and heterogeneity of this cancer type [8,9]. Thanks to these studies, we have now an almost comprehensive catalogue of AML mutated genes. Progress is however hampered as the functional role of these genes in nonpathological conditions is often poorly understood. The prognostic and predictive value of some genetic alterations has recently been reported, revealing the importance of gene mutations in predicting patient clinical parameters, including response to therapy and overall survival [10]. Here we describe the genetic alterations on one of the most frequently mutated gene in AML patients, the “Fms-like tyrosine kinase 3”, FLT3 gene. Different mutations have been identified in this gene and have been associated to a different sensitivity to standard chemotherapy and to FLT3 inhibitor treatments [11,12]. In this review, we summarize our current understanding of the role of different FLT3 mutations in causing drug resistance. We focus on internal tandem duplications (ITDs) occurring in different domains of the FLT3 gene and we describe how the distinct location along the cytoplasmic domain of this transmembrane receptor influences cancer cell sensitivity to drugs. Importantly, the different locations of the ITDs also affect patient prognosis. Thus, the elucidation of the mechanisms underlying FLT3-dependent drug resistance represents a crucial goal with important translational impact on AML patient treatment. In the second part of this report, we describe how mass spectrometry (MS)-based (phospho) proteomics combined with prior knowledge causal networks enable to obtain cell-specific signaling networks and to identify new potential therapeutic targets.

## 2. FLT3

FLT3 belongs to the class III receptor tyrosine kinase (RTKs) group of receptors, characterized by an N-terminal extracellular region consisting of five immunoglobulin-like domains, a juxtamembrane domain (JMD) followed by two tyrosine kinase domains (TKDs). In physiological conditions, FLT3 is mainly expressed in hematopoietic stem cells (HSC) and B-cell progenitors. Its expression decreases during hematopoietic differentiation, with the exception of a subpopulation of monocytes that maintain high levels of FLT3 expression after differentiation [13]. In the absence of ligand, FLT3 is kept inactive by the interaction between the autoinhibitory JM domain and the kinase domain, which blocks the ATP binding site. Activation is mediated by the binding with FL the receptor ligand, which leads to rapid changes in the intracellular domain and to its homodimerization. This conformational change causes the release of the autoinhibitory juxtamembrane domain from the kinase domain and renders the ATP binding site accessible [14]. Activation results in the autophosphorylation of several tyrosine residues (Y589, Y591, Y599). In its active form, FLT3 triggers different signaling pathways, including PI3K/AKT, STAT and MAPK cascades [15], resulting in enhanced proliferation.

### 2.1. FLT3 Mutations in AML

Over the past years, several systems of classification of AML patients have been proposed. The European Leukemia Net (ELN) risk stratification guidelines recommended to stratify patients into three groups with different inferred clinical outcome: favorable, intermediate and adverse. This classification system is based on cytogenetic and molecular criteria [10,16,17]. Alongside the detection of cytogenetic abnormalities at diagnosis, the current World Health Organization (WHO) classification recommends also the assessment of FLT3 mutational status for patients with AML [11] as constitutive activation of the FLT3 kinase is one of the factors associated to poor prognosis of AML patients. FLT3 mutations frequently co-occur with additional genetic alterations. In a recent study, Papaemmanuil et al. sequenced 111 cancer drivers in a cohort of 1540 AML patients enrolled in three independent clinical trials. In Table 1, we report the more frequent genetic alterations co-occurring with FLT3 mutations in this AML patient cohort. Worth to mention, mutation pairs or triplets in most of these patients recapitulate the “two-hit model” proposed by Gilliland and Griffith in 2002. According to this model, AML is the consequence of at least two mutations, each belonging to a different class. Class I mutations confer a proliferative advantage, while class II mutations impair hematopoietic differentiation [8]. As shown in Table 1, FLT3 mutations, which belong to class I, often co-occur with class II mutations, in the DNMT3A, TET2, WT1 and IDH1/2 genes. However, it is also possible to identify genetic associations of FLT3 mutations with other class I genes, such as NRAS, MYC and PTPN11, stressing once more the complexity and heterogeneity of AML.

Among the 1540 AML patients, about 500 carry FLT3 mutations. Indeed, it has been estimated that about 30% of AML patients have their FLT3 gene mutated at diagnosis [11]. FLT3 mutations can be divided into two groups: internal tandem duplications (ITDs) and point mutations (Figure 1). Both these genetic alterations are gain of function mutations, leading to constitutive FLT3 activation and, consequently, aberrant cell proliferation. ITD mutations represent the most common type of FLT3 mutations with a percentage of about 25% of all AML cases [11], while point mutations are identified in 2% to 10% of all AML patients and are localized in the JMD or in the TKDs domains [18,19,20]. Among the different point mutations, the Asp 835 residue in the activation loop is the predominant FLT3 genetic alteration. Interestingly, Kuriyan and colleagues constructed a FLT3 homology model using as a template the structure of c-Kit, which shares 65% sequence identity with the FLT3 kinase domain. Interestingly, they observed that Asp 835, in the model of the active receptor conformation, is located close to a hydrophobic patch. It was suggested that the Asp 835 mutation to a more hydrophobic residue may promote new interactions and stabilize the activation loop in the active conformation [21].

ITDs are characterized by in-frame duplications 3 to 1236 bp in length, mostly in exons 14 and 15, which correspond to the juxtamembrane domain (JMD). The crystal structure of autoinhibited FLT3 leads to direct evidence of how the JMD region may exert its autoinhibitory effect on the catalytic activity of FLT3. More specifically, the insertion of ITD sequences disrupts the autoinhibitory interaction between the juxtamembrane domain and the kinase domain, allowing FLT3 to switch from the inactive to the catalytically active conformation in absence of the FLT3 ligand [14,22]. Such conformational rearrangements cause constitutive autophosphorylation of FLT3 and the consequent phosphorylation of its downstream targets. Interestingly, in 2006 Stirewalt et al., followed in 2009 by Kayser et al., revealed that ITD insertions can also be localized in the TK domain [23,24]. Specifically, in 28.7% of 753 FLT3-ITD-positive AML patients, Breitenbuecher et al. identified an in-frame duplication in the tyrosine kinase domain (TKD) of the FLT3 receptor. ITD insertions were observed in:the TKD1 beta-sheet1 (amino acids 610 to 615) in 24.6% of FLT3-ITD positive AML patientsin the nucleotide binding loop (NBL) (amino acids 616 to 623) in 2% of FLT3-ITD positive AML patientsin the TKD2 beta-sheet2 (amino acids 624 to 630) in 1.3 % of FLT3-ITD positive AML patients.

Interestingly, these observations are clinically relevant as the ITDs in the kinase domain have been associated to unfavorable patient prognosis in terms of complete remission, relapse-free survival and overall survival. It has been observed that the ITD size is also extremely variable from patient to patient and that the ITDs in the JMD and TKD regions are highly variables in length ranging from a few to hundreds of nucleotides [23,24,25,26]. However, the relation, if any, between ITD size and patient prognosis is still poorly understood.

**Figure 1 proteomes-09-00019-f001:**
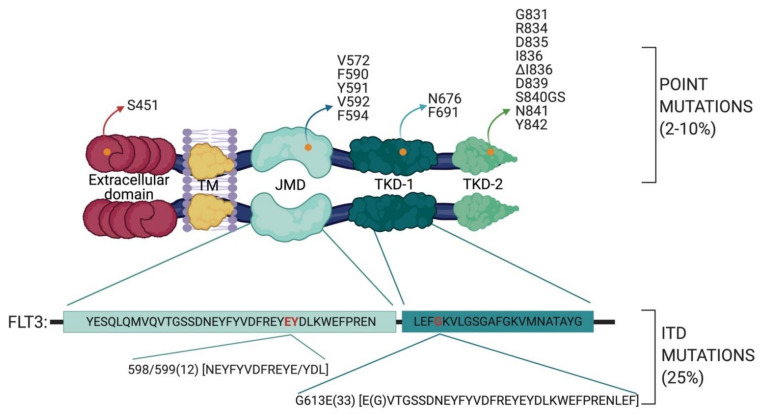
Schematic representation of AML mutational landscape in the FLT3 gene. The FLT3 protein contains 5 functional domains: an immunoglobulin-like extracellular domain, a transmembrane domain (TM), a juxtamembrane domain (JMD) and the two tyrosine kinase domains (TKDs), TKD1 and TKD2. FLT3 point mutations are displayed according to their domain localization (modified from [19,27]). Yellow dots indicate point mutations, whereas amino acid residues in red indicate the residues wherein ITD mutations are frequently located.

Given the importance of FLT3 mutations in AML onset and development, over the past decade, therapeutic protocols based on FLT3 inhibitors have been tested in clinical trials more or less specific kinase inhibitors have been tested either in monotherapy or in combination with chemotherapy. Protocols based on first-generation inhibitors, in monotherapy, are often clinically inefficient possibly because of low potency and selectivity. Thus, these inhibitors are commonly used in combination with standard chemotherapy (e.g., midostaurin plus cytarabine and anthracycline) [27,28]. In recent years, second generation FLT3 inhibitors have been developed. Their enhanced potency and selectivity, as compared to first generation inhibitors promise greater efficacy in specifically targeting the mutated forms of the FLT3 receptor. In Table 2, we have summarized the target specificity of FLT3 inhibitors which are currently under evaluation in clinical trials. 

### 2.2. Chemotherapeutic Resistance Mechanisms in FLT3-Dependent AML

Other than the efforts to develop targeted therapies to block or attenuate the activity of the FLT3 receptor, increasing evidence have shown that cancer cells treated with tyrosine kinase inhibitors (TKIs) tend to acquire additional genetic alterations to escape inhibition. Although some patients obtain great benefit from the addition of FLT3 inhibitors (e.g., midostaurin) to standard chemotherapy [29], a significant percentage of patients experienced leukemia relapse within months after the initial remission [12]. In most cases, relapse is associated to the development of drug resistance. Several distinct mechanisms can lead to drug resistance, including protection of AML cells by the bone marrow environment, evolution and/or expansion of resistant clones and the development of intrinsic, adaptive cellular mechanisms [30]. Interestingly, it has been shown that bone marrow stromal cells can interact with AML cells and determine a different drug response by a FGF receptor dependent mechanism [31]. Our review focuses on cell-autonomous FLT3-driven resistance mechanisms, which can be further classified in primary and secondary resistance. Primary resistance occurs at the onset of the disease, while secondary resistance arises after the early phase of treatment. Among the most frequent innate resistance mechanisms, the co-expression of the wild-type form of FLT3 with the mutant has been associated to a decreased efficacy of combined chemotherapy inhibitor treatment. Indeed, the reduced sensitivity of wild-type FLT3 to TKI treatment accompanied by the chemotherapy-dependent increase of FLT3 ligand promotes downstream proliferative pathways and AML progression. The upregulation and/or increased activity of cytochrome P450 enzymes (CYP3A4) has been shown to increase the drug turnover in FLT3-positive AML as well as in other cancer types [32,33].

Mutations in the FLT3 receptor have been associated with the development of innate or acquired resistance. For example, Linardopoulos’s team generated resistant cells by long-term exposure of FLT3-mutant human cell lines with selective inhibitors (e.g., tandutinib and quizartinib). The resulting resistant cells harbored an additional mutation (D835Y) in the tyrosine kinase domain on the gene with the FLT3-ITD (+) allele. Consistently, the acquired FLT3 D835Y mutation has been recently identified in relapsed FLT3-ITD positive AML patients [34]. These observations indicate that multiple FLT3 mutations can be present in the same patient and can play a role in the modulation of the sensitivity to chemotherapeutic treatments.

Worth to mention, as in other types of leukemia, AML cells harboring pre-existing resistance-mediating mutations can be selected by chemotherapeutic or TKIs treatment. Interestingly, FLT3-ITD mutation has been recognized as a relapse-related genetic marker [30]. Here we focus on the non-canonical ITD mutations characterized by an insertion within the TK domain of FLT3, occurring in about 30% of FLT3-ITD positive AML patients. As already mentioned, these TKD1-ITD FLT3-positive AML patients have a worse prognosis for overall survival and relapse free-survival as compared to AML patients carrying the canonical JMD-ITD mutation [23,24]. We and other demonstrated that, although in cells harboring the two mutations FLT3 activity is turned off upon FLT3 inhibitor treatment, the percentage of TKD1-ITD cells undergoing apoptosis is significantly lower in comparison with cells harboring the canonical ITD mutation in the JM domain (JMD-ITD cells) (Figure 2a). These experiments have been performed by treating patient-derived leukemic cells as well as in stably transfected cell lines with different first and second generation FLT3 inhibitors (e.g., quizartinib and midostaurin) [23,24]. These experiments demonstrated that the different sensitivity of the two ITD mutants cannot be explained by FLT3 inhibitor off-target effects, but it is due to the different location of the ITD in FLT3. In a recent study, the Fischer’s and Heidel’s group(s) aimed at characterizing the impact of FLT3-ITD insertion sites on TKI-therapy in vitro and in vivo. As expected, the retroviral injection of primary murine bone marrow cells harboring either ITD led to lethal myeloproliferation, as previously reported. Their findings show how FLT3 ITD location alone is responsible for a different response to TKI treatment even in AML patients with a complex genetic background [26]. Interestingly, competitive transplantation of JMD- and TKD1-ITDs cells revealed competitive advantage for JMD-ITDs over TKD1-ITDs. This different proliferative capacity does not correlate with drug resistance. As previously reported, cells harboring ITDs within the TKD1 have a significantly decreased sensitivity to TKI treatment in vitro as compared to JMD-ITD cells. This reduced sensitivity of TKD-ITD cells to TKI treatment has been investigated by a genome-wide unbiased analysis of the transcriptome profile. This analysis was performed in 32D-cells infected with viruses expressing different ITD-mutations (three JMD-ITDs and two TKD1-ITDs) as well as in primary patient cells expressing FLT3 JMD and TKD-ITDs (33 FLT3 JMD-ITD patient samples and 16 FLT3 TKD-ITD patient samples). Genes involved in the DNA repair process were found to be differentially modulated in the two ITDs expressing cells. This observation is consistent with the reduced accumulation of γH2AX foci after γ-radiation in TKD1-ITD cells as compared to JMD-ITD cells. Efficient DNA repair in TKD-ITD cells may promote cell survival by protecting cells from cytotoxic treatment.

The reduced sensitivity of TKD-ITD cells is still poorly understood. It is clear, however that the molecular mechanisms underlying the different sensitivity cannot be simply explained by the hyperactivation of the usual culprits, such as AKT, STAT5 and ERK1/2 (Figure 2b). The TKI treatment efficiently blocks FLT3 activation and decreases AKT, STAT5 and ERK1/2 activity in both JMD and TKD-ITD cells [35]. Thus, a simple model whereby constitutive kinase activation results in a linear cascade of signaling events leading to uncontrolled proliferation cannot explain the divergent response to inhibitors observed in AML cells harboring the two ITD mutations. Given the crosstalk and complexity within the cell signaling pathways, it is necessary to consider a scenario where the two FLT3 ITD mutations cause distinct remodeling of the signaling network, which, in turn, mediates a different response to chemotherapy. How are signaling pathways rewired in AML cells harboring the different ITD mutations? To address this question, in the following paragraphs we describe a network-based strategy based on the combination of mass spectrometry-based phosphoproteomics with literature-derived causal networks. 

## 3. A Network-Based Strategy to Revert Chemotherapeutic Resistance in AML

### 3.1. MS-Based Phosphoproteomics

Systematic analysis of signaling network rewiring is essential to decipher the molecular basis of complex biological phenomena, as the different sensitivity of FLT3-ITD positive AML cells. Among the 200 different post-translational modifications, phosphorylation is the most common regulatory mechanisms controlling protein function to transduce signals in cells. In a recent study, it has been estimated that three-fourth of the detected proteome can be phosphorylated [36]. Several high-throughput technologies have been developed to systematically monitor changes in protein phosphorylation. These include reverse phase protein arrays, phospho-specific flow cytometry, mass cytometry and mass spectrometry (MS)-based phosphoproteomics [37]. Advances in sample preparation coupled to developments in hardware [38] and software [39] has, however, given MS-based phosphoproteomics a leading advantage, as this technology offers now the possibility to identify and quantify phosphoproteins and phosphosites in an unbiased, manner with high coverage and accuracy. Protein phosphorylation can be identified and mapped at single amino acid resolution by the analysis of MS spectra because of the characteristic mass shift. In a typical MS-based phosphoproteomic experiment, the procedure can be divided into four steps: (i) sample preparation, including cell fractionation and protein digestion; (ii) phosphopeptide enrichment; (iii) phosphopeptide separation by liquid chromatography (LC) coupled with tandem MS; and (iv) bioinformatic analysis of MS spectra to identify and quantify phosphosites. Historically, MS-based phosphoproteomics have faced many challenges, including the requirement of large amount of starting material and the necessity to fractionate the enriched phosphopeptides. Only a few years ago, the sample preparation workflow was a very complex procedure, technically demanding and applicable to a limited number of experimental conditions [4]. Recently, all these challenges have been addressed and overcome thanks the development of streamline approaches, including the EasyPhos workflow [5]. This method requires a very low amount of starting material and it is easily performed in a single tube in contrast to traditional phosphoproteomic workflows (Figure 3). Briefly, 0.5 to 1 mg of lysate is alkylated and reduced; proteins are digested with trypsin and LysC serine endoproteinase. At this stage it is possible to sample an aliquot of the digested peptides for total proteome analysis. Next, phosphopeptides are enriched by TiO_2_ beads affinity purification (Figure 3 panel 2), eluted, separated using high-pressure liquid chromatography and sprayed in the MS instrument via electrospray. In the mass spectrometer the peptides are first processed in a data-dependent acquisition (DDA) mode, which allows to select the most abundant peptides for fragmentation and identification [6]. The MS spectra are analyzed by different computational software packages. MaxQuant is one of the most frequently used platforms for MS-based proteomics data analysis and enables to obtain information about peptide mass, intensity and presence of specific post-translational modifications [40]. The Andromeda search engine implemented in the MaxQuant environment then allows to search each fragmented peptide and its fragment ion pattern against databases for peptide identification and protein assembly. This step is usually followed by statistical analysis of the resulting dataset and biological interpretation. Proteomics data can be analyzed by the statistical module Perseus of the MaxQuant environment. This software platform allows to analyze large datasets and facilitate their biological and clinical interpretation [41].

This phosphoproteomic workflow can identify up to 10,000 to 20,000 class I phosphorylation sites, which are the ones identified with high confidence (localization probability score > 0.75). The resulting high content phosphoproteomic datasets allow to obtain a comprehensive overview of a cell phosphoprotein profile. Additionally, the described workflow enables to simultaneously process multiple samples, allowing to characterize the cell phosphoproteome in a time and space-resolved manner.

In principle, MS-based phosphoproteomics is the best possible technology to elucidate how signaling pathways are rewired in AML cells harboring the different ITD mutations [42,43,44]. However, it is important to consider that interpreting these large datasets is often not straightforward and simple as one would image and cannot be done manually one-by-one. A number of bioinformatic approaches have been developed to help phosphoproteomic data interpretation [45]. The identification of biological processes differently enriched in lists of modulated phosphosites is a common strategy [46,47]. This analysis is usually accompanied by kinase substrate motifs enrichment analysis [48,49,50], which allows to identify kinases whose activities are likely to be differently modulated in different conditions. In the next paragraphs, we describe how network-based approaches can be applied to extract signaling information from phosphoproteomics dataset, enabling the identification of targets whose modulation is likely to revert the chemotherapeutic resistance of FLT3-positive AML cells.

### 3.2. Integrating Mass-Spectrometry Based Proteomics with Literature-Derived Signaling Networks

Different network-based approaches have been developed to obtain from phosphoproteomic measurements information about signaling pathway modulation in disease. It has emerged over the years that diseases are often caused by genetic alterations that trigger a significant rewiring of the protein interaction network rather than by the simple alteration of the activity of a single protein. Thus, the characterization of a disease state, diagnosis and prognosis requires a proteome wide characterization of protein activities. As phosphorylation is the main post translational mechanism modulating protein activity, MS based phosphoproteomics plays a prominent role in the characterization of the modules of the signaling network that are altered in a disease condition. Phosphoproteomics-based network medicine applies high-throughput phosphorylation profiling in the classification of cancers, therapy planning and prediction of drug response [51]. In a recent study, multilayered proteomic analysis was used to assemble a wide BCR signalosome and new signaling components were identified and validated in HeLa and primary B cells [52] Hijazi et al. applied MS-based phosphoproteomics combined with computational analysis of kinase-phosphosite relationship to decipher the topology of kinase-substrate network. In this study more than 1500 kinase–kinase interactions were predicted and computational strategies were applied to reconstruct a kinase network from phosphoproteomic dataset [53]. Phosphotyrosine enrichment-based phosphoproteomics of 16 AML cell lines led to the characterization of modulated signaling cascades and to the identification of hyperactive kinases as new putative drug targets for AML therapy [54]. Finally, we have recently developed a workflow to overlay -omic data into literature-derived signaling network. This strategy has been applied to elucidate how signaling networks are rewired in different experimental conditions, including breast cancer cells upon metformin treatment, glucose-stimulated pancreatic beta cells and type-2 diabetes islets [55,56,57].

Here we review this approach and discuss its application to elucidate signaling networks rewiring in drug resistant and sensitive FLT3-positive AML cells. The strategy takes advantage of SIGNOR, a manually curated database capturing more than 26,000 causal interactions. Every entry is SIGNOR is a binary, signed and directional interaction between proteins or other biological entities (complexes, chemicals, phenotypes), supported by literature evidence. In addition, entries are linked to additional metadata such the molecular mechanism involved in the regulation (phosphorylation, binding, etc.) and, when available, the amino acid position of post-translational modifications (PTM). At the data of writing SIGNOR stores approximately 9800 regulatory phosphorylation reactions and information for 8800 phosphorylated sites [58,59]. The initial step in the workflow requires the assembly of a human naïve interactome by using the information annotated in SIGNOR capturing experimental evidence of causal interactions reported in the literature irrespective of the cell system or biological context. The second step consists in overlaying onto the network proteomic and phosphoproteomic data, thereby annotating each node with experimental evidence of protein concentration and activity in the context of interest. The result is an AML-specific network where each node is a protein that is expressed in AML cells. By comparing the concentrations and activities of node proteins in different conditions (e.g., drug resistant and non-resistant cells) it is possible to identify subnetworks whose activity is differentially regulated because of differences in protein abundance or activation. 

To reproduce the workflow, the following material is required:(a)Network template: the file should contain a table with at least five columns listing the source nodes, the target nodes, the causal effects (up- or downregulation) and the information about the amino acid position of the phosphorylated site as well as the amino acid sequence context of the phosphosite (sequence window). In our case, this file is the complete list of causal interactions available from the SIGNOR database.(b)Node experimental attributes: a table containing the protein expression levels in specific experimental conditions, as revealed by MS-based proteomic experiments.(c)Edge attributes: a table listing the phosphorylation level of the regulatory phosphopeptides involved in each activation/inactivation reaction, as revealed by the MS-based phosphoproteomic experiment.(d)Cytoscape software installed. For the scope of this example, we used default options. Alternatively, a plethora of adds on applications developed to visualize and analyze networks and omics data are made available at the Cytoscape App store (Table 3).

These are the steps required to obtain a FLT3 specific-causal network.

Download the complete list of interactions from the Download all data section of the SIGNOR database (https://signor.uniroma2.it/downloads.php accessed on 24 April 2021).Upload the complete dataset on the Cytoscape software by setting the columns as follows: ENTITYA>Source Node; ENTITYB>Target Node; TYPEA/B, IDA/B>Source/Target Attribute; EFFECT>Interaction Type; MECHANISM, SEQUENCE, RESIDUE, DIRECT>Interaction Attribute (Figure 4a).Import proteomic data as node attributes.Use the “filter” tab in Cytoscape to select nodes identified in specific experimental conditions as revealed by MS-based proteomic experiments (Figure 4b).Create a subnetwork containing only the selected nodes to obtain a network of proteins expressed in the reference system.Use the “filter” tab in Cytoscape to select nodes with degree (number of connections) ≥ 0 to remove unconnected nodes.Use the “style” tab to modify the layout of the network, e.g., the size of the nodes to reflect protein expression level (Figure 4c).Import phosphoproteomic data as attribute of the edges, using the 15mer sequence in SIGNOR as key (see field SEQUENCE).Use the “style” tab to modify the visual properties of the edges. Use arrow style to show effect and directionality and modify color according to phosphoproteomic data (Figure 4d).

The result of this pipeline is a signed directed graph that includes only proteins (nodes) expressed in a specific context, as per proteomic data, and where edges represent phosphorylation events at specific sites detected by phosphoproteomics.

The resulting network still lacks a crucial piece of information as the activation level of each node is not annotated. Converting phosphorylation data into qualitative information on protein activity is fundamental to interpret signaling network rewiring. Phosphorylation of one or more residues in the activation loop is the most common mechanism to regulate kinase activity. Thus, for many signaling proteins, it is possible to estimate activity from the phosphorylation levels of its regulatory sites. However, there are also pitfalls to consider. Although annotation of the relation between site phosphorylation and kinase activity is annotated in databases such as SIGNOR and PhosphoSitePlus [58,59,60], in MS-based phosphoproteomic datasets only a small percentage (2–3%) of the annotated regulatory phosphosites are identified [55,61]. As a consequence, most of the phosphorylations detected in a typical MS-phosphoproteomic experiment cannot be simply converted into a qualitative estimate of protein activity. Additionally, phosphorylation of different phosphosite in a protein could have opposite consequences on protein activity making the interpretation of phosphorylation experiments even more challenging. Multiple computational strategies have been developed to infer kinase activities from phosphoproteomic datasets [53,62,63,64]. We here discuss a method to address this challenge and obtain a cell specific logic-model that combines the omics derived network with a multiparametric analysis.

**Table 3 proteomes-09-00019-t003:** Cytoscape apps for networks and omics data.

Name	Description
Omnipath App	It allows access to the large collection of network resources of the Omnipath web server. From the 61 web resources the user can import any combination of networks and their respective annotations. The purpose of the app is to link the access to this kind of data to the Cytoscape functionalities [65].
Omics Visualizer	It is a data visualization app; it is ideal for omics data in which each node of the network is associated to multiple values. Indeed, the app allows the user to import files with multiple rows of data for a single node and offers different ways to visualize these data [66].
BiNGO	It is a Cytoscape plug-in of the Biological Networks Gene Ontology resource. It analyzes GO term enrichments and it maps them onto a given network, it uses either the full GO ontologies annotation or the GOSlim ontologies. The annotated graphs generated by BiNGO are flexible and customizable by the standard Cytoscape functionalities [67].
CytoCopteR	It is the graphical interface of CellNOptR. With this App the user can combine literature-derived network with experimental data to build and optimize cell specific and predictive logic networks. It uses different kind of logic formalisms (Boolean steady-state, Boolean multiple steady-state, Boolean time courses through synchronous update, steady-state constrained fuzzy logic and continuous logic-based ODEs) and the user can choose between them depending on the kind and the amount of data to analyze [68].

### 3.3. Optimizing and Building Dynamic Network trough Cell Signaling Experimental Data

The resulting network offers a static snapshot of how the FLT3-ITD mutations impact signaling network and determine a different sensitivity to chemotherapeutic treatments. The goal is that of obtaining a predictive network which can be used as a framework to infer potential therapeutic targets to revert drug resistance. Boolean approaches have been widely applied to obtain context-specific predictive signaling network models [69]. Historically, the first application of logic modeling to signaling pathways dates back to 1969, when Kaufmann used discrete logic to model gene regulation [70]. Next, Huang and Ingber used dynamic Boolean networks to show that specific cell phenotypes (growth, quiescence, differentiation, apoptosis, etc.) might correspond to steady states of the dynamic logic-based model [71]. Over the past decades, logic-based models have been widely used to understand the relationships between signaling network and cell state and to identify promising therapeutic targets. Different computational tools are now available to obtain cell-specific logic models. Here we describe the pipeline to obtain FLT3 ITD-specific logic models by using the freely accessible Cell Network Optimizer (CNO) software, which is also available as a Cytoscape app.

The CNO software takes as input a prior knowledge network (PKN) and a training data set. The PKN could be either downloaded from the many pathway or model databases or curated from causal databases as we have described for the FLT3-ITD specific signaling networks (Figure 4e). The training dataset is a quantitative dataset where the activity of multiple proteins is monitored under different perturbation conditions. First CNO simplify the network through a compression step, including the elimination of non-observable nodes. Next, the software creates a superstructure of Boolean models having all possible logic gates compatible with the compressed graph. At this point, the topology of the compressed network is trained to data through the optimization function [68,72]. At the end of the process the topology of the network is optimized with edges removed or added to improve the ability of the model to reproduce the behavior observed in the training dataset.

To obtain the training dataset, it is necessary to select a panel of proteins, often dubbed “sentinel proteins”, whose activity can be monitored under different perturbation conditions. System perturbations can be obtained by treating cells with small-molecule inhibitors and/or cytokines, whereas the activity of the sentinel proteins is analyzed by experimental approaches such as the Luminex xMAP sandwich assay, which permits to measure multiple protein concentrations or modifications in up to 96 samples in a single run.

In conclusion, this approach enables to obtain dynamic networks describing the different response to perturbations in sensitive and resistant cells.

## 4. Conclusions

Only few studies addressed the impact of FLT3-ITD TKD mutations on patients’ survival and therapy resistance. These studies have contributed to raise new fundamental and applied questions: (i) How do the TKD-ITD mutations alter the structure of the FLT3 receptor? (ii) Can the size and the position of ITD trigger the activation of different signalling pathways? (iii) Can the different ITD localization be considered as a patients’ stratification feature? (iv) Which is the role of co-mutations of AML patients in the TKD-ITD resistance? We have described here a general strategy to address such questions. Understanding the molecular changes occurring in a perturbed system provides the means to build predictive models and to set a rational basis to formulate new hypothesis. This approach can also have a translational impact as the optimized model may be used to test new strategies to revert the drug resistance phenotype in AML ITD patients. AML patients carry a complex mutational landscape which can be explored by taking advantage of the recent technological advances in single-cell techniques. Over the last years, mass cytometry has played a crucial role in the single cell characterization of AML patient-specific signalling [73,74,75,76]. We believe that this is only the first leg of a long journey that in the near future will lead to a comprehensive single-cell proteomic analysis of AML patients and the identification of a personalized therapeutic approach.

## Figures and Tables

**Figure 2 proteomes-09-00019-f002:**
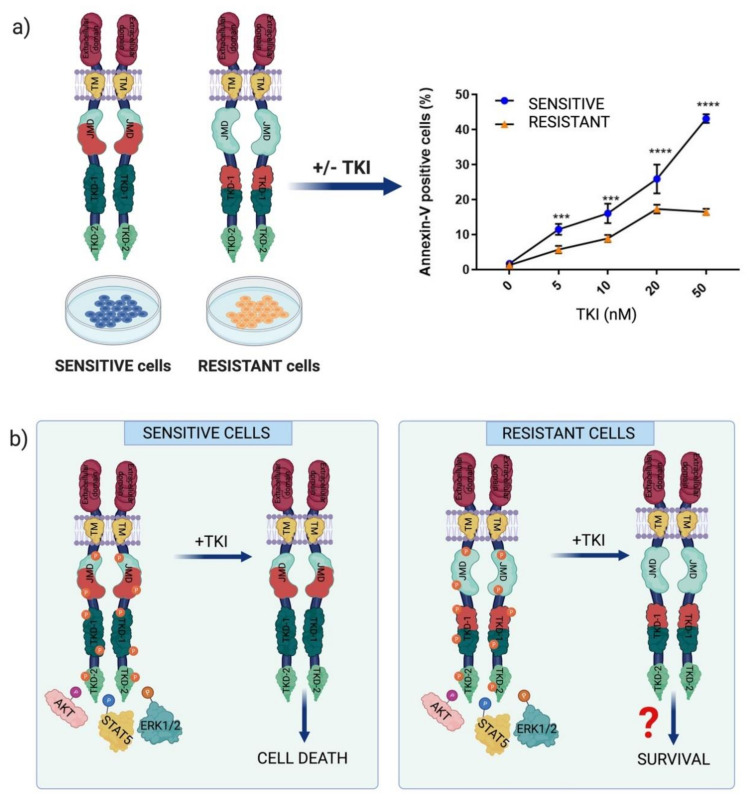
(**a**) Overview of the apoptotic analysis of juxtamembrane domain-internal tandem duplication (JMD-ITD) cells (sensitive cells) and TKD-ITD cells (resistant cells) treated with FLT3 inhibitor (TKI). The graph shows the percentage of Annexin V positive cells (unpublished data). (**b**) Schematic representation of the TKI-response mechanisms in FLT3-ITD positive AML cells.

**Figure 3 proteomes-09-00019-f003:**
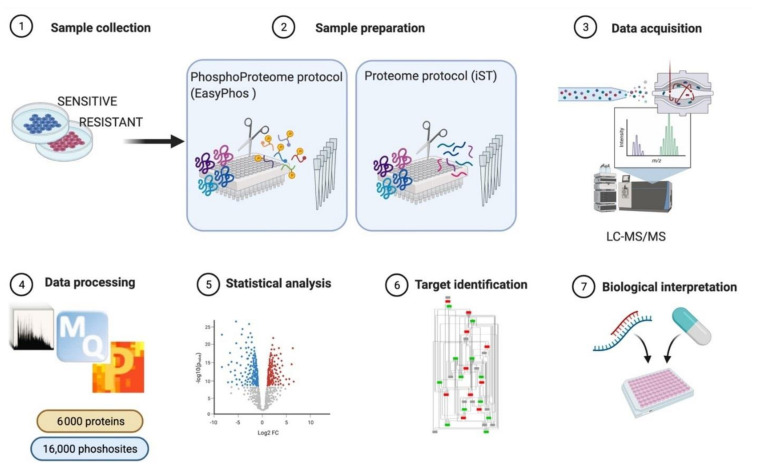
Schematic representation of the experimental strategy applied for the (phospho) proteome analysis of FLT3 sensitive ad resistant cells after tyrosine kinase inhibitor (TKI) treatment.

**Figure 4 proteomes-09-00019-f004:**
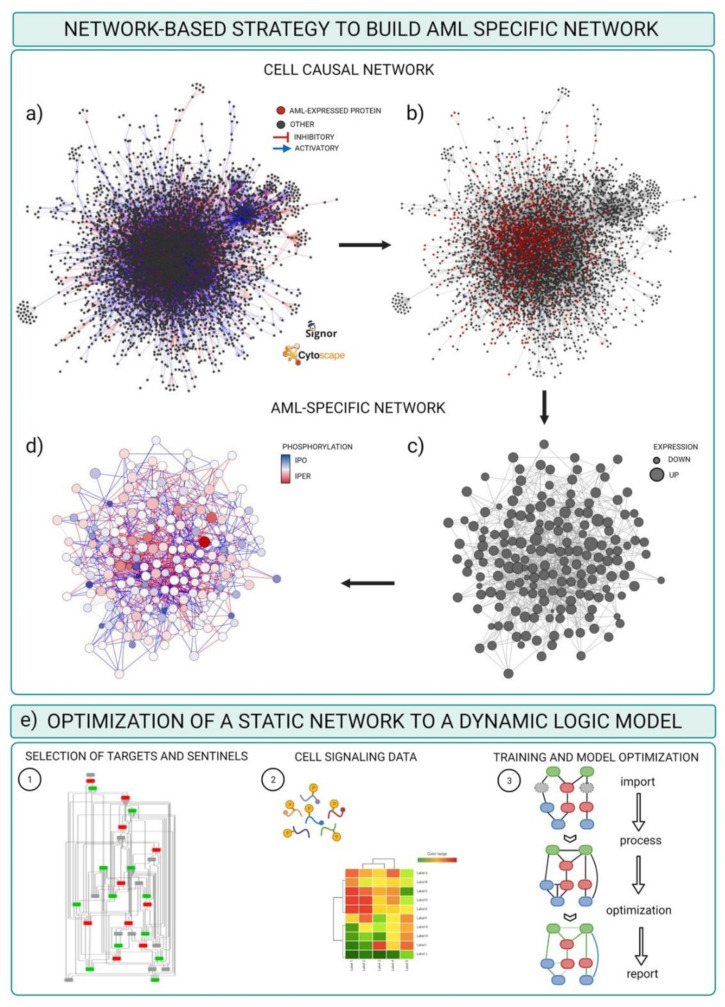
Schematic representation of the pipeline to obtain an AML-specific network. (**a**) Naïve signaling network extracted from SIGNOR database and represented in Cytoscape. All the nodes are dark grey, the edges are red if the interaction is annotated as inhibitory, blue if activatory, light grey if unknown. (**b**) Selection (red nodes) of those nodes associated to at least one quantified value of protein expression, as revealed by MS-based proteomics. (**c**,**d**) AML specific network resulting from the integration of (**a**) with proteomic (**c**) and phospho-proteomic (**d**) data: the size of the nodes represents the level of the protein expression quantified in the proteome; the color of the nodes represents the levels of phosphorylation detected in the phospho-proteome. (**e**) Graphical description of the data training and optimization of the network.

**Table 1 proteomes-09-00019-t001:** List of FLT3 associated genetic alterations identified in 1540 acute myeloid leukemia (AML) patients, as reported by Papaemmanuil et al. [9].

Frequency of FLT3 Co-Occurrent Mutated Genes (*N* = 512)		Frequency of FLT3 Co-Occurrent Mutated Pairs (*N* = 512)	
Gene	*n* of Patients (%)	Gene	*n* of Patients (%)
NPM1	242 (47.3)	NPM1:DNMT3A	130 (25.4)
DNMT3A	168 (32.8)	TET2:NPM1	35 (6.8)
TET2	59 (11.5)	NPM1:IDH1	24 (4.7)
NRAS	51 (9,9)	NPM1:IDH2	24(4.7)
RUNX1	40 (7.8)	NRAS:NPM1	21 (4.1)
WT1	37 (7.2)	PTPN11:NPM1	21 (4.1)
CEBPA	36 (7.0)	RAD21:NPM1	20 (3.9)
MLL	35 (6.8)	TET2:DNMT3A	19 (3.7)
IDH1	34 (6.6)	IDH1:DNMT3A	16 (3.1)
IDH2	33 (6.4)	IDH2:DNMT3A	16 (3.1)
PTPN11	29 (5.7)	MLL:DNMT3A	15 (2.9)
RAD21	29 (5.7)	NRAS:DNMT3A	15 (2.9)
SFRS2	15 (2.9)	RUNX1:DNMT3A	13 (2.5)
MYC	14 (2.7)	WT1:NPM1	13 (2.5)
ASXL1	12 (2.3)	DNMT3A:CEBPA	11 (2.1)
CBL	12 (2.3)	RUNX1:MLL	11(2.1)
EZH2	12 (2.3)	NPM1:MYC	8 (1.6)
KRAS	12 (2.3)	SFRS2:RUNX1	8 (1.6)
PHF6	11 (2.1)	STAG2:NPM1	8 (1.6)
KIT	10 (1.9)	NPM1:KRAS	7 (1.4)
GATA2	9 (1.7)	RAD21: DNMT3A	7 (1.4)
SF3B1	8 (1.6)	TET2:RUNX1	7 (1.4)
MLL2	7 (1.4)	KRAS:DNMT3A	6 (1.2)
TP53	7 (1.4)	PHF6:NPM1	6 (1.2)
U2AF1	7 (1.4)	RUNX1:NRAS	6 (1.2)
NF1	6 (1.2)	TET2:MLL	6 (1.2)
ZRSR2	5 (1)	TET2:PTPN11	6 (1.2)
		NPM1:NF1	5 (0.9)
		NRAS:KRAS	5 (0.9)
		RUNX1:EZH2	5 (0.9)
		STAG2:MLL	5 (0.9)
		TET2:RAD21	5 (0.9)
		TET2:STAG2	5 (0.9)

**Table 2 proteomes-09-00019-t002:** FLT3 inhibitors in clinical trial for AML.

1° Generation					
Inhibitor	Sorafenib	Midostaurin	Sunitinib	Lestaurtinib	Tandutinib
**Target**	FLT3; c-KIT; VEGFR; PDGFR; RAF1	FLT3; c-KIT; PDGFRB; VEGFR	FLT3; c-KIT; KDR; PDGFR	FLT3; JAK2; TRK A	FLT3; PDGFR; c-KIT
**Trial phase**	II/III	III	II	III	I
**FDA approved**	No	Yes	No	No	No
**2° Generation**					
**Inhibitor**	**Quizartinib**	**Gilteritinib**	**Crenolanib**	**Ponatinib**	
**Target**	FLT3; c-KIT; PDGFRa	FLT3; AXL	FLT3; PDGFR	FLT3; BCR-ABL; c-KIT; FGFR1; PDGFRa	
**Trial phase**	III	III	III	I/II	
**FDA approved**	No	Yes	No	No	

FDA: Food and Drug Administration.

## Data Availability

No new data were created or analyzed in this study. Data sharing is not applicable to this article.

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
