# Peer review of "Combining Mass Spectrometry-Based Phosphoproteomics with a Network-Based Approach to Reveal FLT3-Dependent Mechanisms of Chemoresistance"

_proteomes, 2021, doi:10.3390/proteomes9020019_

Round 1

Reviewer 1 Report

Minor

Figure 1: In the scheme of the ITD mutations of the JMD domain, put it in the same color as the tridimensional scheme, as with the TKD-1 domain.

Figure 1-line 161: Bold "landscape in the FLT3 gene."

Line 169: Lowercase "More".

The format of the references is not standardized, in some references the year of publication is not in bold, or even does not appear, as in the case of references 12, 31 or 53.

Reviewer 2 Report

Pugliese et al. submitted the manuscript "Combining Mass Spectrometry-based phosphoproteomics with 2 a network-based approach to reveal FLT3-dependent mecha-3 nisms of chemoresistance.". This is a very comprehensive and well written review. Comprehension of chemoresistance is very challenging and focus on only FLT3 mutated patients is a very interesting approach. 

Major comments:

  1. All TKI inhibitors have different kinome. Gilteritinib inhibits axl for example. How do you explain that what you described was only associated to FLT3 and not to the off target effects?
  2. Chemoresistance could be an intrinsec mechanism or a clonal evolution or selection: could you comment this point?
  3. Patients have usually co mutations, how do you exclude the impact of co mutations?

Reviewer 3 Report

This is a well-written review abounds with sufficient introduction of FLT3 and related proteomics platform. Minor thoughts:

  1. Single-cell proteomics is gaining more popularity whereas its technical challenges have not been well documented so far. Might as well loop in more information on this aspect.
  2. Phosphorylation of FLT3 can vary in different circumstances, from single site to multiple sites. May expand the details and relay them to the corresponding proteomics strategies.

Round 2

Reviewer 2 Report

The authors answered to all comments